# Reverse engineering recurrent networks for sentiment classification reveals line attractor dynamics

**Niru Maheswaranathan**[*]
Google Brain, Google Inc.
Mountain View, CA
nirum@google.com

**Alex H. Williams**[*]
Stanford University
Stanford, CA
ahwillia@stanford.edu

**Matthew D. Golub**
Stanford University
Stanford, CA
mgolub@stanford.edu

**Surya Ganguli**
Stanford and Google Brain, Google Inc.
Stanford, CA
sganguli@stanford.edu

**David Sussillo**
Google Brain, Google Inc.
Mountain View, CA
sussillo@google.com

## Abstract

Recurrent neural networks (RNNs) are a widely used tool for modeling sequential data, yet they are often treated as inscrutable black boxes. Given a trained recurrent network, we would like to reverse engineer it–to obtain a quantitative, interpretable description of how it solves a particular task. Even for simple tasks, a detailed understanding of how recurrent networks work, or a prescription for how to develop such an understanding, remains elusive. In this work, we use tools from dynamical systems analysis to reverse engineer recurrent networks trained to perform sentiment classification, a foundational natural language processing task. Given a trained network, we find fixed points of the recurrent dynamics and linearize the nonlinear system around these fixed points. Despite their theoretical capacity to implement complex, high-dimensional computations, we find that trained networks converge to highly interpretable, low-dimensional representations. In particular, the topological structure of the fixed points and corresponding linearized dynamics reveal an approximate line attractor within the RNN, which we can use to quantitatively understand how the RNN solves the sentiment analysis task. Finally, we find this mechanism present across RNN architectures (including LSTMs, GRUs, and vanilla RNNs) trained on multiple datasets, suggesting that our findings are not unique to a particular architecture or dataset. Overall, these results demonstrate that surprisingly universal and human interpretable computations can arise across a range of recurrent networks.

## 1 Introduction

Recurrent neural networks (RNNs) are a popular tool for sequence modelling tasks. These architectures are thought to learn complex relationships in input sequences, and exploit this structure in a nonlinear fashion. However, RNNs are typically viewed as black boxes, despite considerable interest in better understanding how they function.

Here, we focus on studying how recurrent networks solve document-level sentiment analysis—a simple, but longstanding benchmark task for language modeling [7, 19]. Simple models, such as logistic regression trained on a bag-of-words representation, can achieve good performance in this setting [17]. Nonetheless, baseline models without bi-gram features miss obviously important

---

[*]equal contribution

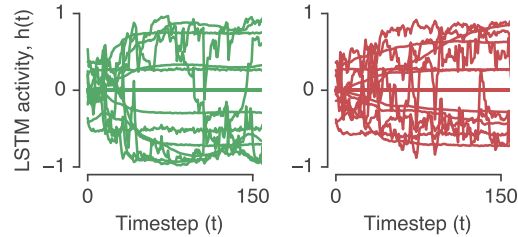

Figure 1: Example LSTM hidden state activity for a network trained on sentiment classification. Each panel shows the evolution of the hidden state for all of the units in the network for positive (left) and negative (right) example documents over the first 150 tokens. At a glance, the activation time series for individual units appear inscrutable.

syntactic relations, such as negation clauses [18]. To capture complex structure in text, especially over long distances, many recent works have investigated a wide variety of feed-forward and recurrent neural network architectures for this task (for a review, see [19]).

We demonstrate that popular RNN architectures, despite having the capacity to implement high-dimensional and nonlinear computations, in practice converge to low-dimensional representations when trained on this task. Moreover, using analysis techniques from dynamical systems theory, we show that locally linear approximations to the nonlinear RNN dynamics are highly interpretable. In particular, they all involve approximate low-dimensional line attractor dynamics–a useful dynamical feature that can be implemented by linear dynamics and can used to store an analog value [13]. Furthermore, we show that this mechanism is surprisingly consistent across a range of RNN architectures. Taken together, these results demonstrate how a remarkably simple operation—linear integration—arises as a universal mechanism in disparate, nonlinear recurrent architectures that solve a real world task.

## 2 Related Work

Several studies have tried to interpret recurrent networks by visualizing the activity of individual RNN units and memory gates during NLP tasks [5, 15]. While some individual RNN state variables appear to encode semantically meaningful features, most units do not have clear interpretations. For example, the hidden states of an LSTM appear extremely complex when performing a task (Fig. 1). Other work has suggested that network units with human interpretable behaviors (e.g. class selectivity) are not more important for network performance [10], and thus our understanding of RNN function may be misled by focusing only on single interpretable units. Instead, this work aims to interpret the entire hidden state to infer computational mechanisms underlying trained RNNs.

Another line of work has developed quantitative methods to identify important words or phrases in an input sequence that influenced the model's ultimate prediction [8, 11]. These approaches can identify interesting salient features in subsets of the inputs, but do not directly shed light into the computational mechanism of RNNs.

## 3 Methods

### 3.1 Preliminaries

We denote the hidden state of a recurrent network at time $t$ as a vector, $\mathbf{h}_t$. Similarly, the input to the network at time $t$ is given by a vector $\mathbf{x}_t$. We use $F$ to denote a function that applies any recurrent network update, i.e. $\mathbf{h}_{t+1} = F(\mathbf{h}_t, \mathbf{x}_t)$.

### 3.2 Training

We trained four RNN architectures–LSTM [4], GRU [1], Update Gate RNN (UGRNN) [2], and standard (vanilla) RNNs–on binary sentiment classifcation tasks. We trained each network type on each of three datasets: the IMDB movie review dataset, which contains 50,000 highly polarized

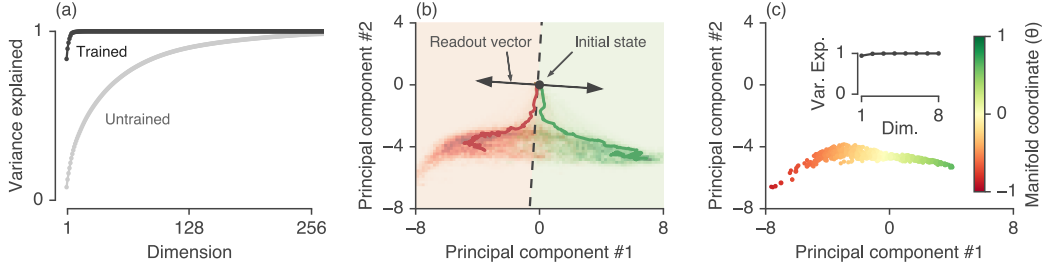

Figure 2: LSTMs trained to identify the sentiment of Yelp reviews explore a low-dimensional volume of state space. (a) PCA on LSTM hidden states - PCA applied to all hidden states visited during 1000 test examples for untrained (light gray) vs. trained (black) LSTMs. After training, most of the variance in LSTM hidden unit activity is captured by a few dimensions. (b) RNN state space - Projection of LSTM hidden unit activity onto the top two principal components (PCs). 2D histogram shows density of visited states for test examples colored for negative (red) and positive (green) reviews. Two example trajectories are shown for a document of each type (red and green solid lines, respectively). The projection of the initial state (black dot) and readout vector (black arrows) in this low-dimensional space are also shown. Dashed black line shows a readout value of 0. (c) Approximate fixed points - Projection of approximate fixed points of the LSTM dynamics (see Methods) onto the top PCs. The fixed points lie along a 1-D manifold (inset shows variance explained by PCA on the approximate fixed points), parameterized by a coordinate $\theta$ (see Methods).

reviews [9]; the Yelp review dataset, which contained 500,000 user reviews [20]; and the Stanford Sentiment Treebank, which contains 11,855 sentences taken from movie reviews [14]. For each task and architecture, we analyzed the best performing networks, selected using a validation set (see Appendix B for test accuracies of the best networks).

## 3.3 Fixed point analysis

We analyzed trained networks by linearizing the dynamics around approximate fixed points. Approximate fixed points are state vectors $\{\mathbf{h}_1^*, \mathbf{h}_2^*, \mathbf{h}_3^*, \cdots\}$ that do not change appreciably under the RNN dynamics with zero inputs: $\mathbf{h}_i^* \approx F(\mathbf{h}_i^*, \mathbf{x}=\mathbf{0})$ [16]. Briefly, we find these fixed points numerically by first defining a loss function $q = \frac{1}{N}\|\mathbf{h} - F(\mathbf{h}, \mathbf{0})\|_2^2$, and then minimizing $q$ with respect to hidden states, $\mathbf{h}$, using standard auto-differentiation methods [3]. We ran this optimization multiple times starting from different initial values of $\mathbf{h}$. These initial conditions were sampled randomly from the distribution of state activations explored by the trained network, which was done to intentionally sample states related to the operation of the RNN.

# 4 Results

For brevity, in what follows we explain our approach using the working example of the LSTM trained on the Yelp dataset (Figs. 2-3). At the end of the results we show a summary figure across a few more architectures and datasets (Fig. 6). We find similar results for all architectures and datasets, as demonstrated by an exhaustive set of figures in the supplementary materials.

## 4.1 RNN dynamics are low-dimensional

As an initial exploratory analysis step, we performed principal components analysis (PCA) on the RNN states concatenated across 1,000 test examples. The top 2-3 PCs explained ∼90% of the variance in hidden state activity (Fig. 2a, black line). The distribution of hidden states visited by untrained networks on the same set of examples was much higher dimensional (Fig. 2a, gray line), suggesting that training the networks stretched the geometry of their representations along a low-dimensional subspace.

We then visualized the RNN dynamics in this low-dimensional space by forming a 2D histogram of the density of RNN states colored by the sentiment label (Fig. 2b), and visualized how RNN states evolved within this low-dimensional space over a full sequence of text (Fig. 2b).

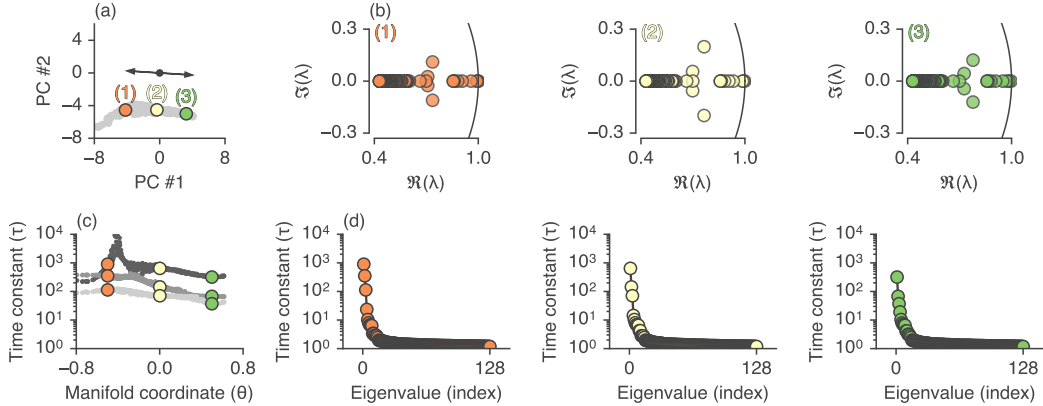

Figure 3: Characterizing the top eigenmodes of each fixed point. (a) Same plot as in Fig. 2c (fixed points are grey), with three example fixed points highlighted. (b) For each of these fixed points, we compute the LSTM Jacobian (see Methods) and show the distribution of eigenvalues (colored circles) in the complex plane (black line is the unit circle). (c-d) The time constants ($\tau$ in terms of # of input tokens, see Appendix C) associated with the eigenvalues. (c) The time constant for the top three modes for all fixed points as function of the position along the line attractor (parameterized by a manifold coordinate, $\theta$). (d) All time constants for all eigenvalues associated with the three highlighted fixed points. The top eigenmode across fixed points has a time constant on the order of hundreds to thousands of tokens.

We observed that the state vector incrementally moved from a central position towards one or another end of the PC-plane, with the direction corresponding either to a positive or negative sentiment prediction. Input words with positive valence ("amazing", "great", etc.) incremented the hidden state towards a positive sentiment prediction, while words with negative valence ("bad", "horrible", etc.) pushed the hidden state in the opposite direction. Neutral words and phrases did not typically exert large effects on the RNN state vector.

These observations are reminiscent of line attractor dynamics. That is, the RNN state vector evolves along a 1D manifold of marginally stable fixed points. Movement along the line is negligible whenever non-informative inputs (i.e. neutral words) are input to the network, whereas when an informative word or phrase (e.g. "delicious" or "mediocre") is encountered, the state vector is pushed towards one or the other end of the manifold. Thus, the model's representation of positive and negative documents gradually separates as evidence is incrementally accumulated.

The hypothesis that RNNs approximate line attractor dynamics makes four specific predictions, which we investigate and confirm in subsequent sections. First, the fixed points form an approximately 1D manifold that is aligned with the readout weights (Section 4.2). Second, all fixed points are attracting and marginally stable. That is, in the absence of input (or, perhaps, if a string of neutral/uninformative words are encountered) the RNN state should rapidly converge to the closest fixed point and then should not change appreciably (Section 4.4). Third, locally around each fixed point, inputs representing positive vs. negative evidence should produce linearly separable effects on the RNN state vector along some dimension (Section 4.5). Finally, these instantaneous effects should be integrated by the recurrent dynamics along the direction of the 1D fixed point manifold (Section 4.5).

## 4.2 RNNs follow a 1D manifold of stable fixed points

The line attractor hypothesis predicts that RNN state vector should rapidly approach a fixed point if no input is delivered to the network. To test this, we initialized the RNN to a random state (chosen uniformly from the distribution of states observed on the test set) and simulated the RNN without any input. In all cases, the normalized velocity of the state vector ($\|\mathbf{h}_{t+1} - \mathbf{h}_t\|/\|\mathbf{h}_t\|$) approached zero within a few steps, and often the initial velocity was small. From this we conclude that the RNN is very often in close proximity to a fixed point during the task.

We numerically identified the location of $\sim$500 RNN fixed points using previously established methods [16, 3]. Briefly, we minimized the quantity $q = \frac{1}{N}\|\mathbf{h} - F(\mathbf{h}, \mathbf{0})\|_2^2$ over the RNN hidden

state vector, $\mathbf{h}$, from many initial conditions drawn to match the distribution of hidden states during training. Critical points of this loss function satisfying $q < 10^{-8}$ were consider fixed points (similar results were observed for different choices of this threshold). For each architecture, we found $\sim 500$ (approximate) fixed points.

We then projected these fixed points into the same low-dimensional space used in Fig. 2b. Although the PCA projection was fit to the RNN hidden states, and not the fixed points, a very high percentage of variance in fixed points was captured by this projection (Fig. 2c, inset), suggesting that the RNN states remain close to the manifold of fixed points. We call the vector that describes the main axis of variation of the 1D manifold $\mathbf{m}$. Consistent with the line attractor hypothesis, the fixed points appeared to be spread along a 1D curve when visualized in PC space, and furthermore the principal direction of this curve was aligned with the readout weights (Fig. 2c).

We further verified that this low-dimensional approximation was accurate by using locally linear embedding (LLE) [12] to parameterize a 1D manifold of fixed points in the raw, high-dimensional data. This provided a scalar coordinate, $\theta_i \in [-1, 1]$, for each fixed point, which was well-matched to the position of the fixed point manifold in PC space (coloring of points in Fig. 2c).

## 4.3 Linear approximations of RNN dynamics

We next aimed to demonstrate that the identified fixed points were marginally stable, and thus could be used to preserve accumulated information from the inputs. To do this, we used a standard linearization procedure [6] to obtain an approximate, but highly interpretable, description of the RNN dynamics near the fixed point manifold. Briefly, given the last state $\mathbf{h}_{t-1}$ and the current input $\mathbf{x}_t$, the approach is to locally approximate the update rule with a first-order Taylor expansion:

$$
\begin{aligned}
\mathbf{h}_t &= F(\mathbf{h}^* + \Delta\mathbf{h}_{t-1}, \mathbf{x}^* + \Delta\mathbf{x}_t) \\
&\approx F(\mathbf{h}^*, \mathbf{x}^*) + \mathbf{J}^{\mathrm{rec}}\Delta\mathbf{h}_{t-1} + \mathbf{J}^{\mathrm{inp}}\Delta\mathbf{x}_t
\end{aligned}
\tag{1}
$$

where $\Delta\mathbf{h}_{t-1} = \mathbf{h}_{t-1} - \mathbf{h}^*$ and $\Delta\mathbf{x}_t = \mathbf{x}_t - \mathbf{x}^*$, and $\{\mathbf{J}^{\mathrm{rec}}, \mathbf{J}^{\mathrm{inp}}\}$ are Jacobian matrices of the system: $J_{ij}^{\mathrm{rec}}(\mathbf{h}^*, \mathbf{x}^*) = \frac{\partial F(\mathbf{h}^*, \mathbf{x}^*)_i}{\partial h_j^*}$ and $J_{ij}^{\mathrm{inp}}(\mathbf{h}^*, \mathbf{x}^*) = \frac{\partial F(\mathbf{h}^*, \mathbf{x}^*)_i}{\partial x_j^*}$.

We choose $\mathbf{h}^*$ to be a numerically identified fixed point and $\mathbf{x}^* = \mathbf{0}^2$, thus we have $F(\mathbf{h}^*, \mathbf{x}^*) \approx \mathbf{h}^*$ and $\Delta\mathbf{x}_t = \mathbf{x}_t$. Under this choice, equation (1) reduces to a discrete-time linear dynamical system:

$$
\Delta\mathbf{h}_t = \mathbf{J}^{\mathrm{rec}}\Delta\mathbf{h}_{t-1} + \mathbf{J}^{\mathrm{inp}}\mathbf{x}_t.
\tag{2}
$$

It is important to note that both Jacobians depend on which fixed point we choose to linearize around, and should thus be thought of as functions of $\mathbf{h}^*$; for notational simplicity we do not denote this dependence explicitly.

By reducing the nonlinear RNN to a linear system, we can analytically estimate the network's response to a sequence of $T$ inputs. In this approximation, the effect of each input $\mathbf{x}_t$ is decoupled from all others; that is, the final state is given by the sum of all individual effects[3].

We can restrict our focus to the effect of a single input, $\mathbf{x}_t$. Let $k = T - t$ be the number of time steps between $\mathbf{x}_t$ and the end of the document. The total effect of $\mathbf{x}_t$ on the final RNN state is $(\mathbf{J}^{\mathrm{rec}})^k \mathbf{J}^{\mathrm{inp}}\mathbf{x}_t$. After substituting the eigendecomposition $\mathbf{J}^{\mathrm{rec}} = \mathbf{R}\mathbf{\Lambda}\mathbf{L}$ for a non-normal matrix, this becomes:

$$
\mathbf{R}\mathbf{\Lambda}^k\mathbf{L}\mathbf{J}^{\mathrm{inp}}\mathbf{x}_t = \sum_{a=1}^{N} \lambda_a^k \mathbf{r}_a \boldsymbol{\ell}_a^\top \mathbf{J}^{\mathrm{inp}}\mathbf{x}_t,
\tag{3}
$$

where $\mathbf{L} = \mathbf{R}^{-1}$, the columns of $\mathbf{R}$ (denoted $\mathbf{r}_a$) contain the *right eigenvectors* of $\mathbf{J}^{\mathrm{rec}}$, the rows of $\mathbf{L}$ (denoted $\boldsymbol{\ell}_a^\top$) contain the *left eigenvectors* of $\mathbf{J}^{\mathrm{rec}}$, and $\mathbf{\Lambda}$ is a diagonal matrix containing complex-valued eigenvalues, $\lambda_1 > \lambda_2 > \ldots > \lambda_N$, which are sorted based on their magnitude.

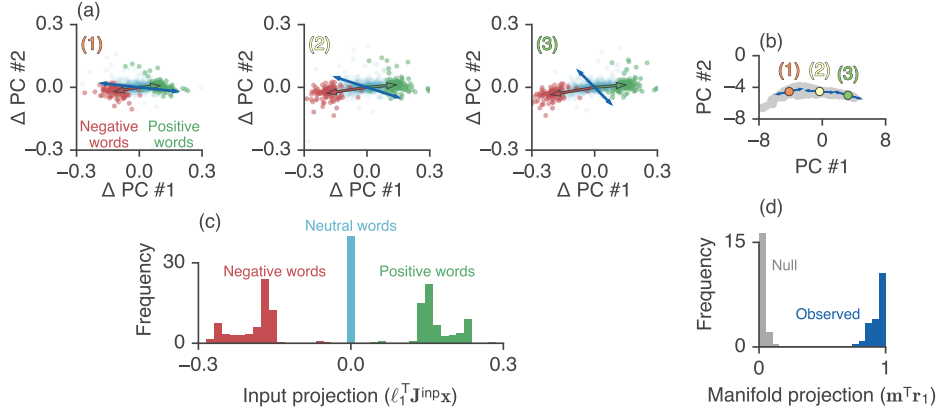

Figure 4: Effect of different word inputs on the LSTM state vector. (a) Effect of word inputs, $\mathbf{J}^{\text{inp}}\mathbf{x}$, for positive, negative, and neutral words (green, red, cyan dots). The green and red arrows point to the center of mass for the positive and negative words, respectively. Blue arrows denote $\boldsymbol{\ell}_1$, the top *left* eigenvector. The PCA projection is the same as Fig. 2c, but centered around each fixed point. Each plot denotes a separate fixed point (labeled in panel b). (b) Same plot as in Fig. 2c, with the three example fixed points in (a) highlighted (the rest of the approximate fixed points are shown in grey). Blue arrows denote $\mathbf{r}_1$, the top *right* eigenvector. In all cases $\mathbf{r}_1$ is aligned with the orientation of the manifold, $\mathbf{m}$, consistent with an approximate line attractor. (c) Average of the projection of inputs with the left eigenvector ($\boldsymbol{\ell}_1^{\top}\mathbf{J}^{\text{inp}}\mathbf{x}$) over 100 positive (green), negative (red), or neutral (cyan) words. Histogram displays the distribution of this input projection over all fixed points. (d) Distribution of $\mathbf{r}_1^{\top}\mathbf{m}$ (overlap of the top right eigenvector with the fixed point manifold) over all fixed points. Null distribution consists of randomly generated unit vectors of the same dimension as the hidden state.

## 4.4 An analysis of integration eigenmodes.

Each mode of the system either reduces to zero or diverges exponentially fast, with a time constant given by: $\tau_a = \left|\frac{1}{\log(|\lambda_a|)}\right|$ (see Appendix C for derivation). This time constant has units of tokens (or, roughly, words) and yields an interpretable number for the effective memory of the system. In practice we find, with high consistency, that nearly all eigenmodes are stable and only a small number cluster around $|\lambda_a| \approx 1$.

Fig. 3 plots the eigenvalues and associated time constants and shows the distribution of all eigenvalues at three representative fixed points along the fixed point manifold (Fig. 3a). In Fig. 3c, we plot the decay time constant of the top three modes; the slowest decaying mode persists after $\sim$1000 time steps, while the next two modes persist after $\sim$100 time steps, with lower modes decaying even faster. Since the average review length for the Yelp dataset is $\sim$175 words, only a small number of modes can retain information from the beginning of the document.

Overall, these eigenvalue spectra are consistent with our observation that RNN states only explore a low-dimensional subspace when performing sentiment classification. RNN activity along the majority of dimensions is associated with fast time constants and is therefore quickly forgotten. While multiple eigenmodes likely contribute to the performance of the network, we restrict this initial study to the slowest mode, for which $\lambda_1 \approx 1$.

## 4.5 Left and right eigenvectors

Restricting our focus to the top eigenmode for simplicity (there may be a few slow modes of integration), the effect of a single input, $\mathbf{x}_t$, on the network activity (eq. 3) becomes: $\mathbf{r}_1\boldsymbol{\ell}_1^{\top}\mathbf{J}^{\text{inp}}\mathbf{x}$. We have dropped the dependence on $t$ since $\lambda_1 \approx 1$, so the effect of $\mathbf{x}$ is largely insensitive to the exact time it was input to system. Using this expression, we separately analyzed the effects of specific words with positive, negative and neutral valences. We defined positive, negative, and neutral words based on the magnitude and sign of the logistic regression coefficients of a bag-of-words classifier.

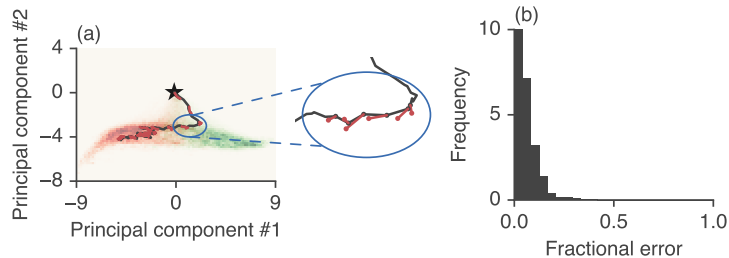

Figure 5: Linearized LSTM dynamics display low fractional error. (a) At every step along a trajectory, we compute the next state using either the full nonlinear system (solid, black) or the linearized system (dashed, red). Inset shows a zoomed in version of the dynamics. (b) Histogram of fractional error of the linearized system over many test examples, evaluated in the high-dimensional state space.

We first examined the term $\mathbf{J}^{\text{inp}}\mathbf{x}$ for various choices of $\mathbf{x}$ (i.e. various word tokens). This quantity represents the instantaneous linear effect of $\mathbf{x}$ on the RNN state vector. We projected the resulting vectors onto the same low-dimensional subspace shown in Fig. 2c. We see that positive and negative valence words push the hidden state in opposite directions. Neutral words, in contrast, exert much smaller effects on the RNN state (Fig 4).

While $\mathbf{J}^{\text{inp}}\mathbf{x}$ represents the instantaneous effect of a word, only the features of this input that overlap with the top few eigenmodes are reliably remembered by the network. The scalar quantity $\boldsymbol{\ell}_1^\top \mathbf{J}^{\text{inp}}\mathbf{x}$, which we call the *input projection*, captures the magnitude of change induced by $\mathbf{x}$ along the eigenmode associated with the longest timescale. Again we observe that the valence of $\mathbf{x}$ strongly correlates with this quantity: neutral words have an input projection near zero while positive and negative words produced larger magnitude responses of opposite sign. Furthermore, this is reliably observed across all fixed points. Fig. 4c shows the average input projection for positive, negative, and neutral words; the histogram summarizes these effects across all fixed points along the line attractor.

Finally, if the input projection onto the top eigenmode is non-negligible, then the right eigenvector $\mathbf{r}_1$ (which is normalized to unit length) represents the direction along which $\mathbf{x}$ is integrated. If the RNN implements an approximate line attractor, then $\mathbf{r}_1$ (and potentially other slow modes) should align with the principal direction of the manifold of fixed points, $\mathbf{m}$. In essence, this prediction states that an informative input pushes the current RNN state along the fixed point manifold and towards a neighboring fixed point, with the direction of this movement determined by word or phrase valence. We indeed observe a high degree of overlap between $\mathbf{r}_1$ and $\mathbf{m}$ both visually in PC space (Fig. 4b) and quantitatively across all fixed points (Fig. 4d).

### 4.6 Linearized dynamics approximate the nonlinear system

To verify that the linearized dynamics (2) well approximate the nonlinear system, we compared hidden state trajectories of the full, nonlinear RNN to the linearized dynamics. That is, at each step, we computed the next hidden state using the nonlinear LSTM update equations ($\mathbf{h}_{t+1}^{\text{LSTM}} = F(\mathbf{h}_t, \mathbf{x}_t)$), and the linear approximation of the dynamics at the nearest fixed point ($\mathbf{h}_{t+1}^{\text{lin}} = \mathbf{h}^* + \mathbf{J}^{\text{rec}}(\mathbf{h}^*)(\mathbf{h}_t - \mathbf{h}^*) + \mathbf{J}^{\text{inp}}(\mathbf{h}^*)\mathbf{x}_t$). Fig. 5a shows the true, nonlinear trajectory (solid black line) as well as the linear approximations at every point along the trajectory (red dashed line). To summarize the error across many examples, we computed the relative error $\|\mathbf{h}_{t+1}^{\text{LSTM}} - \mathbf{h}_{t+1}^{\text{lin}}\|_2 / \|\mathbf{h}_{t+1}^{\text{LSTM}}\|_2$. Fig. 5b shows that this error is small (around 10%) across many test examples.

Note that this error is the single-step error, computed by running either the nonlinear or linear dynamics forward for one time step. If we run the dynamics for many time steps, we find that small errors in the linearized system accumulate thus causing the trajectories to diverge. This suggests that we cannot, in practice, replace the full nonlinear LSTM with a *single* linearized version.

### 4.7 Universal mechanisms across architectures and datasets

Empirically, we investigated whether the mechanisms identified in the LSTM (line attractor dynamics) were present not only for other network architectures but also for networks trained on other datasets

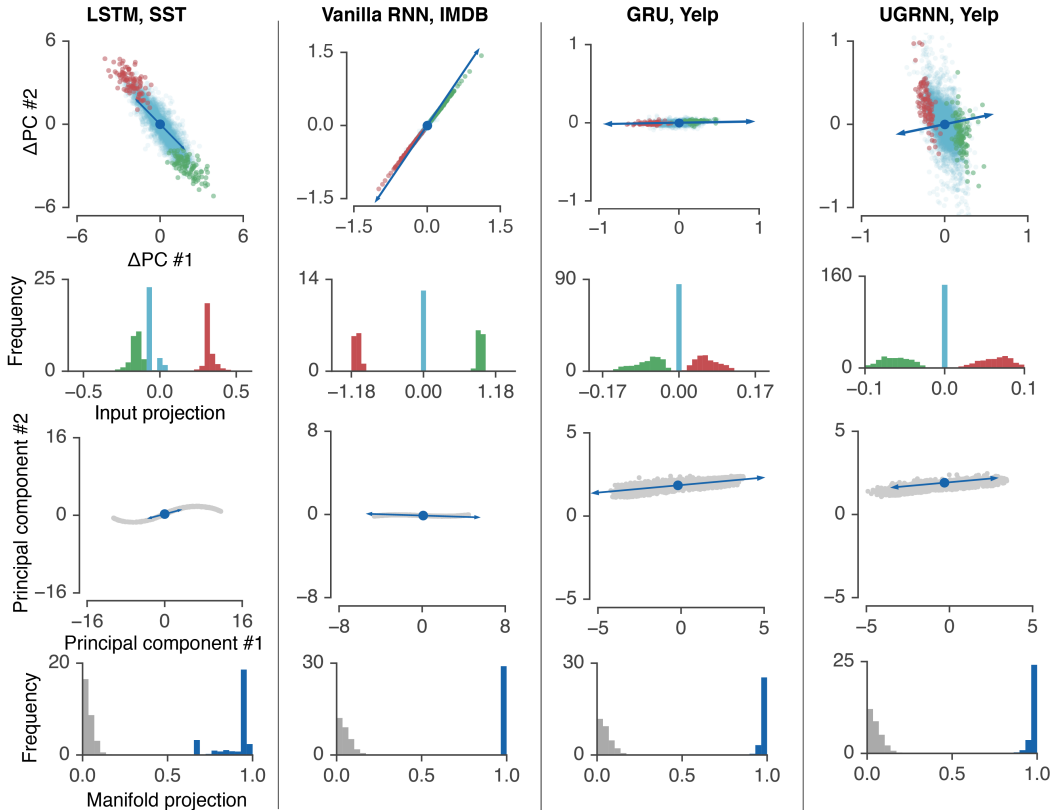

Figure 6: Universal mechanisms across architectures and datasets (see Appendix A for all other architecture-dataset combinations). Top row: comparison of left eigenvector (blue) against instantaneous effect of word input $\mathbf{J}^{\text{inp}}\mathbf{x}$ by valence (green and red dots are positive and negative words, compare to Fig. 4a) for an example fixed point. Second row: Histogram of input projections summarizing the effect of input across fixed points (average of $\boldsymbol{\ell}_1^{\top}\mathbf{J}^{\text{inp}}\mathbf{x}$, compare to Fig. 4c). Third row: Example fixed point (blue) shown on top of the manifold of fixed points (gray) projected into the principal components of hidden state activity, along with the corresponding top right eigenvector (compare to Fig. 4b). Bottom row: Distribution of projections of the top right eigenvector onto the manifold across fixed points (distribution of $\mathbf{r}_1^{\top}\mathbf{m}$, compare to Fig. 4d).

used for sentiment classification. Remarkably, we see a surprising near-universality across networks (but see Supp. Mat. for another solution for the VRNN). Fig. 6 shows, for different architectures and datasets, the correlation of the the top left eigenvectors with the instantaneous input for a given fixed point (first row), as well as a histogram over the same quantity over fixed points (second row). We observe the same configuration of a line attractor of approximate fixed points, and show an example fixed point and right eigenvector highlighted (third row) along with a summary of the projection of the top right eigenvector along the manifold across fixed points (bottom row). We see that regardless of architecture or dataset, each network approximately solves the task using the same mechanism.

## 5 Discussion

In this work we applied dynamical systems analysis to understand how RNNs solve sentiment analysis. We found a simple mechanism—integration along a line attractor—present in multiple architectures trained to different sentiment analysis tasks. Overall, this work provides preliminary, but optimistic, evidence that different, highly intricate network models can converge to similar solutions that may be reduced and understood by human practitioners.

In summary, we found that in nearly all cases the key activity performed by the RNN for sentiment analysis is simply counting the number of positive and negative words used. More precisely, a slow mode of a local linear system aligns its left eigenvector with the current effective input, which itself

nicely separates positive and negative word tokens. The associated right eigenvector then represents that input in a direction aligned to a line attractor, which in turn is aligned to the readout vector. As the RNN iterates over a document, integration of negative and positive words moves the system state along this line attractor, corresponding to accumulation of evidence by the RNN towards a prediction.

Such a mechanism is consistent with a solution that does not make use of word order when making a decision. As such, it is likely that we have not understood all the dynamics relevant in the computation of sentiment analysis. For example, we speculate there may be some yet unknown mechanism that detects simple bi-gram negations of one word by another, e.g. "not bad," since the gated RNNs performed a few percentage points better than the bag-of-words model. Nonetheless, it appears that approximate line attractor dynamics represent a fundamental computational mechanism in these RNNs, which can be built upon by future investigations.

When we compare the overall classification accuracy of the Jacobian linearized version of the LSTM with the full nonlinear LSTM, we find that the linearized version is much worse, presumably due to small errors in the linear approximation that accrue as the network processes a document. Note that if we directly train a linear model (as opposed to linearizing a nonlinear model), the performance is quite high (only around 3% worse than the LSTM), which suggests that the error of the Jacobian linearized model has to do with errors in the approximation, not from having less expressive power.

We showed that similar dynamical features occur in 4 different architectures, the LSTM, GRU, UGRNN and vanilla RNNs (Fig. 6 and Supp. Mat.) and across three datasets. These rather different architectures all implemented the solution to sentiment analysis in a highly similar way. This hints at a surprising notion of universality of mechanism in disparate RNN architectures.

While our results pertain to a specific task, sentiment analysis is nevertheless representative of a larger set of modeling tasks that require integrating both relevant and irrelevant information over long sequences of symbols. Thus, it is possible that the uncovered mechanisms—namely, approximate line attractor dynamics—will arise in other practical settings, though perhaps employed in different ways on a per-task basis.

## Acknowledgments

The authors would like to thank Peter Liu, Been Kim, and Michael C. Mozer for helpful feedback and discussions.

## Footnotes

[2]We also tried linearizing around the average embedding over all words; this did not change the results. The average embedding is very close to the zeros vector (the norm of the difference between the two is less than $8 \times 10^{-3}$), so it is not surprising that using that as the linearization point yields similar results.

[3]We consider the case where the network has closely converged to a fixed point, so that $\mathbf{h}_0 = \mathbf{h}^*$ and thus $\Delta\mathbf{h}_0 = \mathbf{0}$.

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
