[Supplementary Material · sentiment_camera_ready_SUPP.pdf]

# A   Additional figures

Below we provide figures summarizing the linear integration mechanism for each combination of architecture (LSTMs, GRUs, Update Gate RNNs, Vanilla RNNs) and dataset (Yelp, IMDB, and Stanford Sentiment). Note that the first figure, LSTMs trained on Yelp, reproduces the figures in the main text–we include it here for completeness. The description of each panel is given in Figure 7, note that these descriptions are the same across all figures. We find that these mechanisms are remarkably consistent across architectures and datasets.

Figure 7:  Summary plots for LSTM on Yelp reviews. **(upper left)** PCA on RNN hidden states - PCA applied to all hidden states visited during 1000 test examples for untrained (light gray) vs. trained (black) LSTMs. After training, most of the variance in LSTM hidden unit activity is captured by a few dimensions. **(upper middle left)** RNN state space - Projection of RNN hidden unit activity onto the top two principal components (PCs). 2D histogram shows density of visited states for test examples colored for negative (red) and positive (green) documents. Star indicates the initial hidden state. **(upper middle right)** Approximate fixed points - Projection of approximate fixed points of the RNN dynamics (see Methods) onto the top PCs. The fixed points lie along a 1-D manifold, parameterized by a coordinate $\theta$ (see Methods). **(upper right)** Time constant ($\tau$) of memory as a function of position along the line attractor, $\theta$. **(lower left)** Instantaneous effect of word inputs, $\mathbf{J}^{\text{inp}}\mathbf{x}$, for positive (green), negative (red), and neutral (cyan) words. Blue arrows denote $\boldsymbol{\ell}_1$, the top *left* eigenvector. The PCA projection is the same as Fig. 2c, but centered around each fixed point. **(lower middle left)** Average of $\boldsymbol{\ell}_1^\top \mathbf{J}^{\text{inp}}\mathbf{x}$ over 100 different words, shown for positive, negative, neutral words. **(lower middle right)** Same plot as in Fig. 2c, with an example fixed point highlighted (approximate fixed points in grey). Blue arrows denote $\mathbf{r}_1$, the top *right* eigenvector. **(lower right)** Distribution of $\mathbf{r}_1^\top \mathbf{m}$ (overlap of the top right eigenvector with the fixed point manifold) over all fixed points. Null distribution is randomly generated unit vectors of the size of the hidden state.

Figure 8: Summary plots for GRU on Yelp reviews. See first supplemental figure (LSTM on Yelp) for description.

Figure 9: Summary plots for UGRNN on Yelp reviews. See first supplemental figure (LSTM on Yelp) for description.

Figure 10: Summary plots for VRNN on Yelp reviews. See first supplemental figure (LSTM on Yelp) for description. Note this VRNN has an unstable oscillation as shown in distribution of hidden states in upper middle left PCA plot.

Figure 11: Summary plots for LSTM on IMDB reviews. See first supplemental figure (LSTM on Yelp) for description.

Figure 12: Summary plots for GRU on IMDB reviews. See first supplemental figure (LSTM on Yelp) for description.

Figure 13: Summary plots for UGRNN on IMDB reviews. See first supplemental figure (LSTM on Yelp) for description.

Figure 14: Summary plots for VRNN on IMDB reviews. See first supplemental figure (LSTM on Yelp) for description.

Figure 15: Summary plots for LSTM on SST reviews. See first supplemental figure (LSTM on Yelp) for description.

Figure 16: Summary plots for GRU on SST reviews. See first supplemental figure (LSTM on Yelp) for description.

Figure 17: Summary plots for UGRNN on SST reviews. See first supplemental figure (LSTM on Yelp) for description.

Figure 18: Summary plots for VRNN on SST reviews. See first supplemental figure (LSTM on Yelp) for description.

## B    Additional methods

Table 1: Test accuracies across all RNN architectures and datasets.

|  | Bag of words | Vanilla RNN | Update Gate RNN | GRU | LSTM |
|---|---|---|---|---|---|
| Yelp 2015 | 93.37% | 92.96% | 95.67% | 95.84% | 95.05% |
| IMDB | 88.53% | 87.08% | 87.96% | 86.86% | 86.93% |
| Stanford Sentiment | 79.74% | 78.09% | 77.74% | 80.25% | 80.09% |

## C    Integration timescales

In order to convert the eigenvalue, $\lambda$, for a particular mode of the dynamical system into a timescale in terms of tokens, we do the following.

First, we note that in the linearized system, the dynamics of the hidden state projected onto a paritcular eigenvector (with corresponding eigenvalue $\lambda$) are given by $h(t) = \lambda h(t-1)$. Therefore, we have $h(t) = \lambda^t h(0)$. We would like to relate this eigenvalue to the corresponding time constant ($\tau$) when we write the solution as an exponential: $h(t) = h(0)e^{-t/\tau}$. Equating the two yields an expression for the time constant in terms of the eigenvalue:

$$h(0)\lambda^t = h(0)e^{-t/\tau}$$
$$\lambda = e^{-1/\tau}$$
$$\tau = \left| \frac{1}{\ln|\lambda|} \right|,$$

where we introduce the absolute value as we define the timescale to be non-negative.