[Reviews · NeurIPS 2019]

Reviewer 1



UPDATE after reading author rebuttal: Look forward to the changes in the final version of the paper. Detailed comments: 1. Understanding of RNNs for sentiment classification task - theoretical analysis backed by empirical observations: This work takes up the sentiment classification task. This work figured out some fixed points and centered their analysis of RNNs around them. The RNN states can be cast into a 1-dimensional manifold of these fixed points. The PCA of RNN states across examples reveal that training helps RNNs figure out a lower-dimensional representation. Interestingly the movement along this low dimensional manifold is minimal in absence of inputs or presence of neutral/un-informative words, whereas they show more movements if polarity bearing words are present, thus, showing linear separability effects along this 1-D manifold. Further analysis using eigenvalue decomposition also support the low-dimensional manifold argument. However, the authors also state that approximation of full non-linear LSTM by linear dynamics is not the best idea in the long run even though for some steps the approximation may be close enough. 2. Theoretically and empirically, the paper shows that even the complex dynamics of RNNs can be approximated by linear and low-dimensional dynamics which are more human interpretable: The presence of low-dimensional manifold around fixed points has been backed by multiple theoretical analysis and empirical observations : (1) PCA of RNN states, (2) Eigen value decomposition and through figures 2 and 3. Moreover, the linear separability depending on polarity bearing words has been demonstrated through Figure 4. Figure 6 show consistency across different variants of RNNs on different datasets. A very positive thing about this paper is the authors limit themselves in overstating the significance about their contributions and pave the way for future research work by (a) pointing out negated polarity bearing 'not bad' may not be perfectly understood by this framework (b) sentiment classification is a particular representative task and it remains to be seen how this mechanism work for other settings. 3. Some critical comments: a) The paper is not very well-written and a little hard to read. It would be better to structure the paper with a 'preliminary' section before Section 3. There it would be better to set the notations and definitions like fixed points, linear attractor dynamics, etc. for more novice readers. b) Many of the mathematical results (for example eq 3) could be structured as theorem/lemma/result. c) Many mathematical observations are simply stated inside paragraphs and get lost, some degree of highlighting will be essential to get the main message out. d) The figures could be better explained in relation to the mathematical results including more details. Overall my scores are as below(out of 10): Originality: 8 Quality: 7 Clarity: 5 Significance: 8

Reviewer 2



Although this paper relies heavily on the well-established field of dynamic systems analysis, I found this paper to be refreshingly innovative in that it takes these tools an applies them in a new way. This is a refreshingly research focused paper compared to the other papers I've reviewed for NeurIPS this year. I like that the authors tackle a difficult problem, that of interpretability of RNN's, in a very principles manner. The paper's quality is good in that it explores a variety of important avenues (e.g., multiple data sets, multiple models, multiple measures of the dynamics, etc.) The paper is very clear and easy to read. I comment the authors for this. As for significance, interpretability is of key relevance in academia, business, and government. This paper provides a new lens for looking at this problem.

Reviewer 3



POST REBUTTAL UPDATE: The authors answered my concerns, and I'm increasing the score to 8. The authors train RNNs on a basic NLP task – sentiment classification. They then use dynamical systems tools to show that the network implements this as a line attractor – perhaps the simplest model of evidence accumulation. Every word is projected onto the line attractor according to its valence, and moves the dynamics towards the correct decision. This mechanism was shown in tasks that were neuroscience-inspired [1], and it’s an important contribution to show that it also arises in tasks that are “pure” machine learning. Major comments: 1. There is inherent variability in the dynamical objects observed: a. Different architectures have different input projection separation (LSTM on IMDB for instance). b. Different points on the line attractor have different q values (not shown, but likely given prior work [2], [3]) c. Different points on the line attractor have different time constants (Fig. 3c for instance) d. Different points on the line attractor have different linearized dynamics error (Fig. 5b) All this variability can be harnessed to try and understand which factors contribute to performance[3]. For instance, If the drift is suddenly larger – do you see that evidence accumulates faster at these points? 2. Bigrams are mentioned, but not analyzed. It could be that this analysis is complex, or the results are inconclusive. But this should be reported. At the very least, show what happens in the dynamical level for the expression “not bad”. Minor comments: 3. Appendix A2 shows that bag of words is not always worse than trained RNNS. (line 238-239) 4. Figure 1 is not clear. Is this an individual neuron for many documents? Many neurons for one document? 5. Section 3.1 – add a reference to appendix A2 6. Line 111 “no input”. Is the natural choice zero input, or the average of all neutral words, or average of all words? 7. Line 114 – Figure 1D does not exist 8. 123: “that THAT the” 9. LSTM vs. VRNN on SST seem to show an opposite trend in their performance compared to their input projections. [1] V. Mante, D. Sussillo, K. V. Shenoy, and W. T. Newsome, “Context-dependent computation by recurrent dynamics in prefrontal cortex,” Nature, vol. 503, no. 7474, pp. 78–84, Nov. 2013. [2] D. Sussillo and O. Barak, “Opening the Black Box: Low-dimensional dynamics in high-dimensional recurrent neural networks,” Neural Comput., vol. 25, no. 3, pp. 626–649, 2013. [3] D. Haviv, A. Rivkind, and O. Barak, “Understanding and Controlling Memory in Recurrent Neural Networks,” ArXiv190207275 Cs Stat, Feb. 2019.

[Author Response · NeurIPS 2019]

We wish to thank all of the reviewers for their time and thorough reading of our paper! Specific concerns are addressed:

**Reviewer #1** We appreciate the reviewer's suggestions regarding clarity. To improve this, we have: **(1)** Added a preliminaries section which introduces our mathematical notation; **(2)** Highlighted key results and observations that were previously buried stated inside paragraphs in sections 4.2, 4.4, and 4.5; and **(3)** Added more detail to the captions in Figures 1, 3, and 4. We chose not to restructure the mathematical results as theorem/lemma/result (as suggested by the reviewer) as we felt that most of our mathematical statements were largely definitions (such as eq. (3)).

**Reviewer #2** Addressing the suggested improvements: **(1)** We have added the suggested summary sentence "the key activity performed by the RNN for sentiment analysis is simply counting the number of positive and negative words used" to the discussion. **(2)** We started with binary sentiment classification, but are actively working on more tasks. For multi-level sentiment classification (e.g. 5-way), our hypothesis is that the networks will still use a 1D line attractor, but that this attractor will be curved such that different readouts will partition different sections of the line attractor (corresponding to different levels of evidence), but still yielding low-dimensional dynamics. We are currently running this experiment and will include its results in the final version of the paper (likely in the supplement, due to space constraints). **(3)** The classification accuracy of the Jacobian linearized model is much worse than the LSTM, due to small errors in the linear approximation that accrue as the network processes a document. Note that if we directly train a linear model, the performance is quite high (only around 3% worse than the LSTM), which suggests that the error of the linearized model has to do with errors in the approximation, not from having less expressive power. We have included a few sentences about this in the discussion. **(4)** We have added a derivation of the expression relating the eigenvalue ($\lambda$) to the time constant ($\tau$) in the supplement, along with a corresponding reference in the main text.

**Reviewer #3** **(Major point 1.)** We agree with the reviewer that a systematic study of the variability we see in the dynamical structures in our analysis is warranted. Assessing if and how this variability is related to performance differences is something we wish to pursue in future work. We have begun some of these investigations, and have found that the small differences in drift and Q values do not seem to affect the performance (their values are too small to have an effect over typical document lengths in these datasets). **(Major point 2.)** As mentioned in the discussion, we have yet to systematically analyze negation bigrams. We have done some preliminary analysis (see Figure 1, at right) which suggests that RNNs are capable of correctly accounting for 'not' tokens. We have a few ideas for how to uncover these mechanisms (e.g. using switched linear approximations), however, this remains as future work. **(Minor point 4.)** Added more detail to the Figure 1 caption, explaining that it is many neurons for one document. **(Minor points 5. and 3.)** Added a reference to the accuracy table in the appendix in Section 3.1. The bag of words does have performance close to that of RNNs, especially for smaller datasets (providing further support for using linear approximations of the RNNs). **(Minor point 6.)** Regarding the input point around which we linearize: in the paper, we linearized around zero input. We also tried linearizing around the average embedding of all words, this

Figure 1: Probing the RNN with negation bigrams. Projection of RNN hidden states onto the top two PCs for two different input sequences that differ only by two tokens (replacing '**and very**' with '**but not**' in the middle of the sequence). The trajectories start out the same as the initial tokens are identical. They then diverge at the critical tokens, moving in opposite directions along the readout (the readout is aligned with the y-axis; not shown). After these two tokens, the rest of the sequence is also identical (tokens not shown to remove clutter). Note how the presence of the negation bigram changes the effect of future tokens on the hidden state.

does not change the results (indeed, the average embedding of all words is very close to the zeros vector—the norm of the average embedding is $7.6 \times 10^{-3}$). We have added a footnote noting this in the main text. **(Minor point 7.)** Removed the incorrect reference to Fig. 1D. **(Minor point 8.)** Fixed typo. **(Minor point 9.)** We have not correlated the performance with things like the input projections. The projection histograms are over the top positive and negative words, whereas the performance (over test examples) depends on the particular words that show up in those examples; as such, it may not make sense to correlate them.

[Meta-Review · NeurIPS 2019]

This paper provides insightful analysis into what decision processes are actually implemented by a trained recurrent network for sentiment classification, and uncover simple line attractor dynamics. All reviewers agree that this is interesting and illuminating, and that this work shows a good example of what can be done to open the black box of deep systems.